# Outer membrane protein N expressed in Gram-negative bacterial strain of *Escherichia coli* BL21 (DE3) Omp8 Rosetta strains under osmoregulation by salts, sugars, and pHs

**Watcharin Chumjan**⬤*, **Akira Sangchalee, Cholthicha Somwang, Nattida Mookda, Sriwannee Yaikeaw, La-or Somsakeesit**⬤

Department of Chemistry, Rajamangala University of Technology Isan Khon Kaen Campus, Khon Kaen, Thailand

* chumjanw@gmail.com, watcharin.cu@rmuti.ac.th

## Abstract

This study presented the expression of the outer membrane protein N in *E. coli* BL21 (DE3) Omp8 Rosetta under its growth condition and by osmoregulation. The effects of osmotic stress caused by salts, sugars, or pH values on the survival of the target Gram-negative bacterial strain of *E. coli* BL21 (DE3) Omp8 Rosetta and OmpN expression remain unknown. Here, tryptone yeast extract with varied salts and concentrations was initially used to generate an LB broth medium. To show how salts and concentration affect bacterial growth, the optical density at 600 nm was measured. The findings supported the hypothesis that salts and concentrations control bacterial growth. Moreover, a Western blotting study revealed that OmpN overexpression was present in all tested salts after stimulation with both glucose and fructose after being treated individually with anti-OmpN and anti-histidine tag polyclonal antibodies on transferred nitrocellulose membrane containing crude OmpN. Following the presence of the plasmid pET21b(+)/*ompN*-BOR into *E. coli* BL21 (DE3) Omp8 Rosetta, which was expressed in the recombinant OmpN protein (BOR), OmpN expression was demonstrated for all monovalent cations as well as $MgCl_2$. All of the tested salts, except for $BaCl_2$, $NaH_2PO_4$, and $KH_2PO_4$, showed overexpression of recombinant BOR after Isopropyl β-D-1-thiogalactopyranoside (IPTG) induction. Using $CH_3COONa$, both with and without IPTG induction, there was very little bacterial growth and no OmpN expression. With NaCl, a pH value of 7 was suitable for bacterial development, whereas KCl required a pH value of 8. According to this research, bacterial growth in addition to salts, sugars, and pH values influences how the OmpN protein is produced.

## Introduction

To measure the development of bacteria in their growth environment, salt is typically used to make a variety of tryptone-yeast media preparations, known as a Luria Bertani broth (LB broth) [1–7]. Studies using monovalent and divalent cations have been carried out both in

**Data Availability Statement:** All relevant data are within the paper and its Supporting Information files.

**Funding:** This research project was supported by the Rajamangala University of Technology Isan (Contract no. ENG21/65) and Thailand Science Research and Innovation (TSRI) (Contract no. FF66-P1-024). WC was funded by the Rajamangala University of Technology Isan (Contract no. ENG21/65), Thailand Science Research and Innovation (TSRI) (Contract no. FF66-P1-024). The funders had no role in study design, data collection and analysis, decision to publish, or preparation of the manuscript.

**Competing interests:** The author have declared that no competing interest exist.

vivo and in vitro in order to assess bacterial growth and determine the ion conductance of numerous outer membrane proteins [8–16]. Previous studies have revealed that the influence of the anion considerably outweighs that of the cation. The order of the effects of the anion was as follows: $F > SO_4 > PO_4 > Cl > NO_3 > Br > I > CNS$, whereas the order of the cation was typically $Ca > Sr > Mg > Cs > Rb > K > Na > Li$ [17]. Another outcome was that sodium and potassium chloride did not inhibit the *E. coli* bacterium, while calcium and iron salts did. The best concentration of bact. coli growth was found at 0.20 M, regardless of the NaCl content (0.05, 0.10, 0.20, 0.30, or 0.40 M) or different pH values [18]. The results demonstrated the accelerating action of the salts on an increase in the speed of organism growth when compared to a medium containing 1 per cent peptone with either 0.2 M NaCl or 0.1 M $Na_2SO_4$ [19].

Typically, *Escherichia coli* models for bacterial growth and the generation of outer membrane proteins are derived from gram-negative bacterial strains. *E. coli* DH5α and *E. coli* BL21 (DE3) Omp8 Rosetta are the two major strains of *E. coli* used for cloning and the production of outer membrane proteins [20–25]. Each is a distinct bacterial strain. BL21 (DE3) Omp8 Rosetta is *E. coli* $B^E$, whereas *E. coli* DH5α is *E. coli* K-12. In addition, all outer membrane proteins are expressed in *E. coli* BL21 (DE3) Omp8 Rosetta, with the exception of LamB, OmpF, OmpC, and OmpA because it lacks the genes required for producing those porins [26]. Outer membrane protein N, as well as OmpF, OmpC, and PhoE, were first found and identified in *E. coli* K-12. Except for PhoE, they are all cation selective [27]. Both the ion conductance and cation selectivity of OmpF were investigated using black lipid membranes. OmpF channels preferred the permeation of alkali metal ions over chloride, according to selectivity tests at low concentrations, which also indicated a preference for smaller cations. These findings imply that the permeating cation and charged residues lining the channel walls interact explicitly [10]. Chemically changing OmpF to $Ca^{2+}$ selectivity led to a reduction in pore conductance, which was another conclusion. At $Ca^{2+}$ mole fractions of $< 10^{-3}$, the conductance of glutathione-modified LECE decreased. Zero-current potential tests showed that the selectivity of $Ca^{2+}$ versus monovalent cations was the highest in the presence of $Li^+$ and the lowest in the presence of $Cs^+$. Thus, it could be illustrated that the LECE pore was sufficiently sized to permit the passage of monovalent cations after $Ca^{2+}$ had been bound [8].

One of the reasons for the production of outer membrane proteins is osmotic stress caused by salts, sugars, pH value, and temperature, which activates the gene involved in the expression of those proteins [28,29]. In *Klebsiella pneumoniae*, for example, high osmotic stress causes the KbvR to be upregulated, which increases the expression of OmpK36 [30]. Moreover, higher levels of osmotic stress caused by NaCl impact OmpC expression increases *Escherichia coli*'s kanamycin resistance [31,32]. A hyperosmotic adaptation at basic pH value was developed by *E. coli* wild type, which had both the *ompF* and *ompC* genes [33]. Herein, the authors chose and used the *E. coli* BL21 (DE3) Omp8 Rosetta strain as a model to investigate the effects of salts that obtain the monovalent and divalent cations in the medium as well as to monitor bacterial growth and OmpN expression levels. To create the tryptone-medium preparation, various salts were used. To monitor bacterial growth, the optical density was measured at 600 nm. Additionally, a Western blotting analysis of OmpN protein expression was carried out.

## Materials and methods

### Bacterial strains, sugars, chemicals, LB broth preparations, and instruments

The *E. coli* BL21 (DE3) Omp8 Rosetta bacterial strains were given by Dr Wipa Suginta, a professor at the Vidyasirimedhi Institute of Science and Technology (VISTEC) in Rayong, Thailand. A 1,000 ml solution of $dH_2O$, 10 g of tryptone, 10 g of sodium chloride (NaCl), and 5 g of

yeast extract were used to make a LB broth medium with a pH value of 6.5. Also, an alternative LB broth medium that had different salts was developed, and the pH values were presented (S1 and S2 Tables). All chemicals including tryptone, yeast extract, glucose, fructose, lithium chloride (LiCl), potassium chloride (KCl), rubidium chloride (RbCl), caesium chloride (CsCl), magnesium chloride ($MgCl_2$), barium chloride ($BaCl_2$), calcium chloride ($CaCl_2$), ammonium chloride ($NH_4Cl$), sodium acetate ($CH_3COONa$), disodium phosphate ($Na_2HPO_4$), sodium phosphate ($NaH_2PO_4$) and potassium phosphate ($KH_2PO_4$) were purchased from Titan Biotech Ltd., India, Vivantis, Malaysia and Sigma-Aldrich, Denmark. At the Department of Chemistry, Faculty of Engineering, Rajamangala University of Technology Isan, Khon Kaen campus, the optical density at 600 nm was measured using a spectrophotometer (Cecil EC1011, 1,000 series).

## Bacterial growth at varied concentrations and different salts

The bacterial cell culture was made by mixing 30 per cent glycerol stock (2–5 μl) of a bacterial strain of *E. coli* BL21 (DE3) Omp8 Rosetta into a 50 ml LB broth medium and incubated at 37˚C with regular shaking for 18 hours at 200 rpm (overnight culture). Numerous studies have demonstrated how the presence of NaCl in the medium affects bacterial growth [5]. Additionally, in vitro investigations utilizing black lipid membrane analyses have shown that various salts also impact the ion conductance of the outer membrane proteins [10,14]. Thus, the author used various salts (such as LiCl, NaCl, KCl, RbCl, CsCl, $MgCl_2$, $BaCl_2$, $CaCl_2$, $NH_4Cl$, $CH_3COONa$, $Na_2HPO_4$, $NaH_2PO_4$ and $KH_2PO_4$) to design and prepare the LB broth media at various concentrations (0, 0.025, 0.05, 0.075, 0.1, 0.25, 0.5, 0.75, 1, 1.5 and 2 M) and also measured the pH values of the medium. The incubated overnight culture was then pipetted and directly transferred to each of the test tubes containing the salts mentioned above, with a final optical density of 600 nm ($OD_{600nm}$) equal to 0.1. All test tubes containing bacteria were shaken continuously at 37˚C for 18 hours. Bacterial growth was then assessed using a spectrometer at $OD_{600nm}$. Another starter culture left by assessment using a spectrometer was then directly added into 5 ml of LB broth medium containing various salts and concentrations until a final $OD_{600nm}$ equalled 0.1, and after which all test tubes were shaken continuously at 200 rpm (at 37˚C) until the $OD_{600nm}$ reached 0.5–0.7. Subsequently, each tube was filled with 125 μl of 1 M glucose and fructose to final concentration of 0.025 M and followed by shaking at 200 rpm for 3 hours. OmpN expression was evaluated on a 12 per cent SDS-PAGE gel for 60 minutes at 120 volts. Bacterial cells containing OmpN protein were employed for Western blot detection.

In addition, pH value and temperature tests were carried out. Briefly, LB broth media containing 0.25 M NaCl and KCl were made separately at different pH values, ranging from pH 4 to 10, with citric acid or Tris-base adjustments. The *E. coli* BL21 (DE3) Omp8 Rosetta starter culture was used to prepare the competent cells, and the recombinant plasmid (pET21b (+)/*ompN*) was transformed and selected in LB broth agar medium containing ampicillin (100 μg/μl) and kanamycin (25 μg/μl). After incubation at 37˚C for 18 hours, the colonies were picked up and then grown in LB broth media (~25 ml) containing ampicillin (100 μg/μl), kanamycin (25 μg/μl) and 1% glucose at any pH value (including pH 4, 5, 6, 7, 8, 9, and 10) and shaken at 200 rpm for 18 hours at 37˚C. After that, the starter culture was then transferred to a fresh LB broth medium (4 ml) containing ampicillin (100 μg/μl) and kanamycin (25 μg/μl) and an $OD_{600nm}$ equal to 0.01. After shaking all the test tubes containing bacterial strains at 200 rpm for 4 hours at 37˚C until the $OD_{600nm}$ reached up 0.6–0.7, 0.5 mM IPTG was added directly to each test tube and shaken continuously at 200 rpm and a temperature of 37˚C for 3 hours. The recombinant OmpN protein expression was then separated on a 12 per cent SDS-PAGE gel at 120 volts for 60 minutes, followed by Western blotting detection.

For temperature effect on OmpN expression, *E. coli* BL21 (DE3) Omp8 Rosetta harboring recombinant plasmid (pET21b(+)/*ompN*) was grown in a LB broth medium containing ampicillin (100 μg/μl), kanamycin (25 μg/μl) and 1% glucose and shaken at 200 rpm for 18 hours at 37˚C. The starter culture was then transferred to a fresh LB broth medium (4 ml) containing ampicillin (100 μg/μl), kanamycin (25 μg/μl) and an $OD_{600nm}$ equal to 0.01. All test tubes containing bacterial strains were shaken at 200 rpm and 37˚C until the $OD_{600nm}$ reached 0.6–0.7, 0.5 mM IPTG was added to each tube separately and then shaken continuously at 200 rpm for 3 hours at 20, 25, 30, 35, and 37˚C. The recombinant OmpN protein expression was then analysed on a 12 per cent SDS-PAGE gel at 120 volts for 60 minutes followed by Western blotting detection.

## Confirmation of OmpN expression level by Western blotting analysis

The *E. coli* BL21 (DE3) Omp8 Rosetta strains containing the OmpN and recombinant OmpN protein were separated on a 12 per cent SDS-PAGE gel. Using a semi-dry gel blotting technique, the protein band from an SDS-PAGE gel was transferred to a nitrocellulose membrane (Amersham, Biosciences). The transferred membrane was then incubated overnight at 4˚C in a blocking solution containing 1x PBS (pH 7.4) and 5% skim milk (non-fat). It was then incubated for 1 hour at room temperature the following day. The membrane was incubated with anti-OmpN and anti-histidine tag polyclonal antibody dilution (1:20,000) in 2 per cent skim milk, 1x PBS (pH 7.4). After incubation with antiserum, the membrane was washed five times in 1x PBS containing 0.1 per cent Tween 20 (PBS-T), then incubated for 1 hour at room temperature in a 1:20,000 dilution of goat anti-rabbit IgG (HRP) in 1x PBS containing 2 per cent skim milk, 1x PBS (pH 7.4). The membrane was then washed three times in PBS-T and twice in 1x PBS for a total of five minutes per wash. The chemiluminescence detection stage was carried out by incubating the membrane for 3 minutes at room temperature with a 1:1 ratio of chemiluminescence substrate. In a dark room, the membrane was wrapped in saran wrap and exposed to an X-ray film.

## Statistical analysis

Each test was performed at least three times, and the mean and standard error of each test were revealed. The statistical evaluation was done using GraphPad Prism 5.0 (GraphPad Software, La Jolla, CA, USA). A statistical analysis that compared groups of tests for bacterial growth, such as *E. coli* BL21 (DE3) Omp8 Rosetta strains grown in tryptone-yeast medium prepared with various salts supplement versus no salts addition in tryptone-yeast medium as a control, was done using the unpaired t-test. ***P <0.001 was used to evaluate statistical significance.

## Results

### The effect of salts and concentrations on the growth of Gram-negative bacterial growth

Herein, we described the *E. coli* BL21 (DE3) Omp8 Rosetta bacterial strain, which typically expressed OmpN, our target outer membrane protein N, in the outer membrane. The first experiment showed bacterial growth at an optical density of 600 nm at varying salt concentrations. On LB broth media containing LiCl, NaCl, KCl, RbCl, and CsCl, respectively, the bacterial strains were cultivated (Fig 1A). With the exception of CsCl, all salts displayed strong growth at 0.25 M, the optimum concentration. Bacterial growth was inhibited at both the highest and lowest dosages. With no additional salt, no bacterial growth was observed.

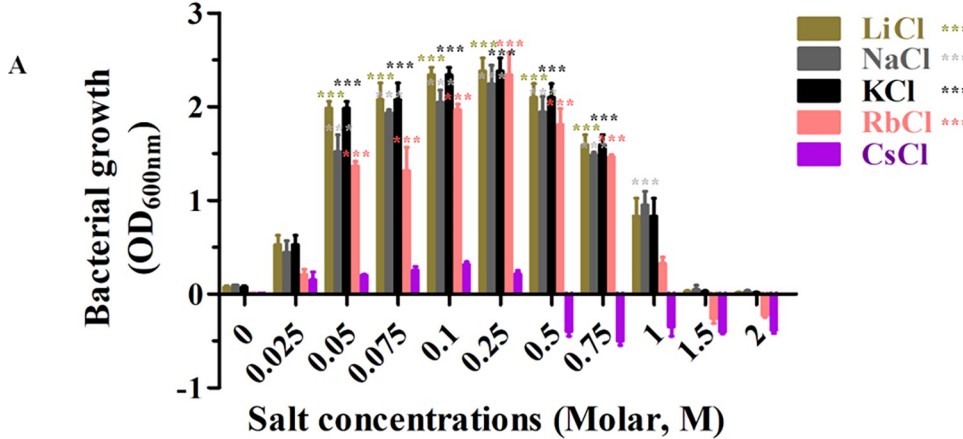

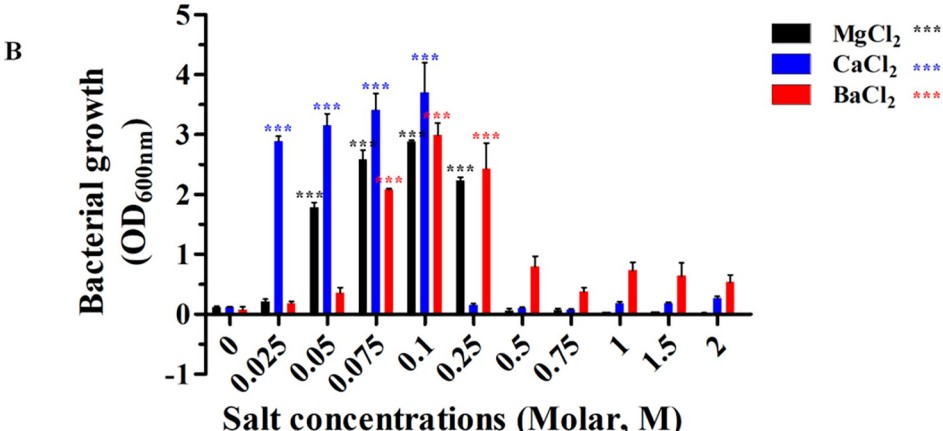

**Fig 1. *coli* BL21 (DE3) Omp8 Rosetta strain bacterial growth at optical density 600 nm in LB broth with salts at different concentrations (0–2.0 M). *E*.** LiCl, NaCl, KCl, RbCl, CsCl (A), and $MgCl_2$, $CaCl_2$, and $BaCl_2$ (B) are all ingredients in LB broth. It is important to note that all of the data came from three different experiments.

Additionally, $MgCl_2$, $CaCl_2$, and $BaCl_2$ were added to the LB broth medium to screen the bacterial growth (Fig 1B). The growth was reduced at 0.25 M concentration, but showed strong results at 0.1 M and an optical density of 600 nm. Bacterial growth in LB broth media containing either $CaCl_2$ or KCl was compared (S1 Fig). The bacterial strain started to grow in the presence of $CaCl_2$ at 0.025 M to 0.1 M but was inhibited from 0.25 M to 2.0 M. When compared to KCl at 0.1 M, $CaCl_2$ had a stronger impact on bacterial growth at 600 nm.

We also tested the impact of ammonium ions from $NH_4Cl$ and $CH_3COONH_4$. $NH_4Cl$ at concentrations between 0.025 and 1.0 M was found to promote bacterial growth, while $CH_3COONH_4$ did not. In comparison to $CH_3COONa$, $NH_4Cl$ increased bacterial growth considerably at 0.25 M (Fig 2A). We also showed how the phosphates $KH_2PO_4$, $NaH_2PO_4$, and $Na_2HPO_4$ affected bacterial growth at 600 nm (Fig 2B). Bacterial growth was achieved by these salts at concentrations between 0.025 M and 0.25 M. In comparison to the addition of $NaH_2PO_4$ and $Na_2HPO_4$ to LB broth media, $KH_2PO_4$ showed greater bacterial growth at 0.1 M. With the exception of CsCl, $CaCl_2$, $CH_3COONH_4$, and $NaH_2PO_4$, all salts significantly

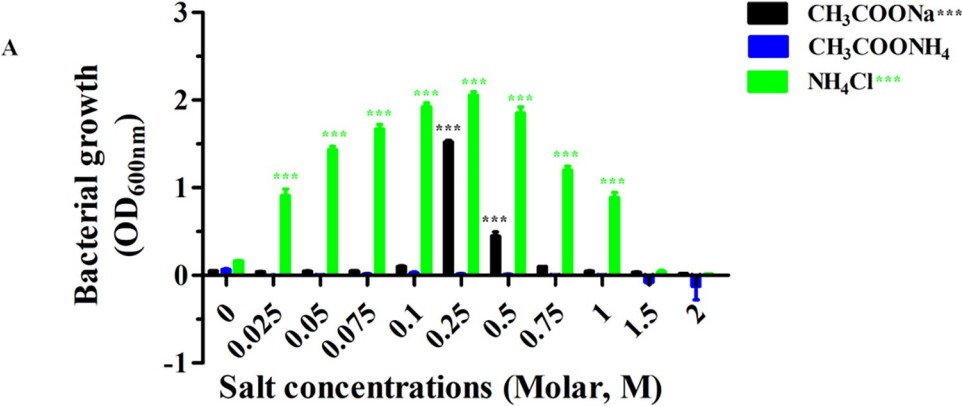

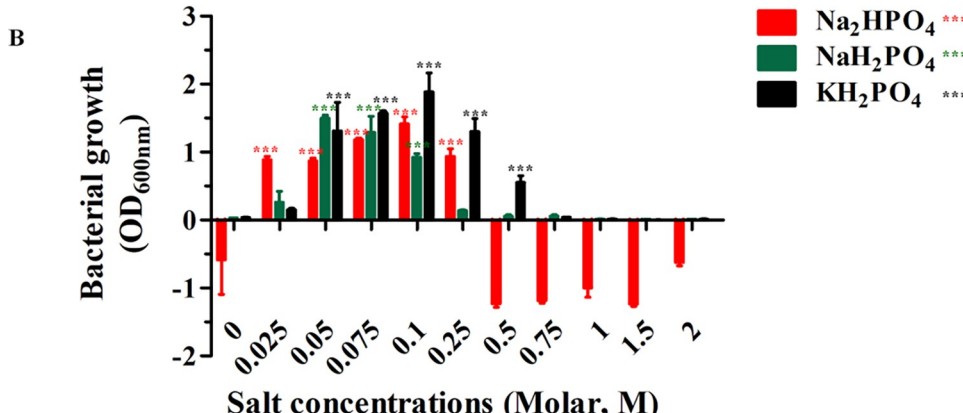

**Fig 2. coli BL21 (DE3) Omp8 Rosetta strain bacterial growth at optical density 600 nm in LB broth with salts at different concentrations (0–2.0 M). E.** $CH_3COONa$, $CH_3COONH_4$, and $NH_4Cl$ (A), and $Na_2HPO_4$, $NaH_2PO_4$, and $KH_2PO_4$ (B). It is important to note that all of the data came from three different experiments.

influenced the growth of *E. coli* BL21 (DE3) Omp8 Rosetta when their effects on bacterial growth were tested at 0.25 M (Fig 3A). After these bacterial strains were grown in either CsCl or $CH_3COONH_4$, no bacterial growth was observed. All salts, especially $CaCl_2$, demonstrated a significant effect on bacterial growth (Fig 3B). After the bacteria were cultivated for 18 hours at 37°C, the increased pH value of the medium was observed (Table 1). The findings indicated that the *E. coli* BL21 (DE3) Omp8 Rosetta strains produced ammonia ($NH_3$) as a by-product of the metabolism of carbon and nitrogen sources for the bacteria to live and raise the pH value of the LB broth. High pH value was observed for KCl during its optimal growth concentration range between 0.025 M and 0.75 M because it accelerated this bacterial growth (Table 2).

## OmpN protein expression with and without induction of mono-sugars on Gram-negative bacterial strain of *E. coli* BL21 (DE3) Omp8 Rosetta

A Western blot was used to discern the level of OmpN protein expression using an anti-OmpN polyclonal antibody based on the bacterial growth profiles. At four hours, the *E. coli* BL21 (DE3) Omp8 Rosetta starting culture was transferred into various salts with a growth

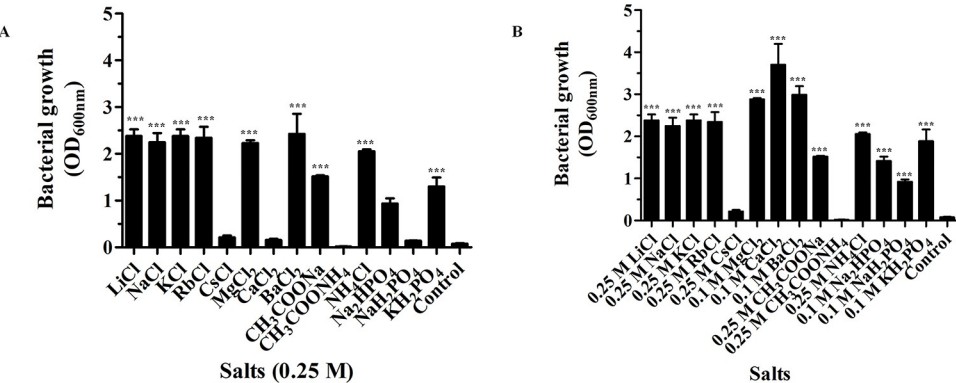

**Fig 3.** *coli* BL21 (DE3) Omp8 Rosetta strain bacterial growth at optical density 600 nm was observed in LB broth medium. *E.* With all salts at 0.25 M (A) and at the optimum growth concentration (B). ***P<0.001 relative to control (at 0 M). It is important to note that all of the data came from three different experiments.

concentration such as 0.25 M for LiCl, NaCl, KCl, RbCl, $CH_3COONa$ and $NH_4Cl$; 0.1 M for $MgCl_2$, $BaCl_2$ and $CaCl_2$; 0.075 M for $Na_2HPO_4$, $NaH_2PO_4$ and $KH_2PO_4$ (Fig 4A). In comparison to the positive control (LB broth medium containing 0.17 M NaCl), $CH_3COONa$ caused a reduction in bacterial growth. A Western blotting analysis using an anti-OmpN polyclonal antibody revealed low OmpN expression in the absence of sugar stimulation. When a bacterial strain was cultivated in LB broth media with either $NaH_2PO_4$ or $Na_2HPO_4$, a high expression level was observed (Fig 4B). The nitrocellulose membrane was stained with Ponceau S to allow

**Table 1. The pH values displayed before and after *E. coli* BL21 (DE3) Omp8 Rosetta addition and cultured in LB broth containing salts at the optimum growth concentration at 37°C, 200 rpm for 18 hours.**

| Salts | Salt concentrations (Molar, M) | The pH values (±SD) | |
|---|---|---|---|
| | | *E. coli* BL21 (DE3) Omp8 Rosetta's addition | |
| | | - | + (37°C, 200 rpm, 18 hours) |
| Control | 0 | 6.30 ±0.03 | 6.07 ±0.02 |
| LiCl | 0.25 | 6.14 ±0.03 | 8.05 ±0.04 |
| NaCl | 0.25 | 6.26 | 8.21 ±0.01 |
| KCl | 0.25 | 6.25 ±0.02 | 8.31 ±0.01 |
| RbCl | 0.25 | 6.45 ±0.20 | 7.58 ±0.29 |
| CsCl | 0.25 | 6.52 ±0.09 | 7.31 ±0.11 |
| $MgCl_2$ | 0.1 | 6.03 ±0.01 | 7.76 ±0.01 |
| $CaCl_2$ | 0.1 | 5.81 ±0.04 | 6.58 ±0.12 |
| $BaCl_2$ | 0.1 | 5.97 ±0.02 | 7.38 ±0.01 |
| $NH_4Cl$ | 0.25 | 6.31 ±0.01 | 7.96 |
| $CH_3COONa$ | 0.25 | 6.94 ±0.01 | 6.82 ±0.01 |
| $CH_3COONH_4$ | 0.25 | 7.31 ±0.05 | 7.27 ±0.03 |
| $Na_2HPO_4$ | 0.075 | 7.09 ±0.01 | 8.22 ±0.02 |
| $NaH_2PO_4$ | 0.075 | 5.93 ±0.02 | 8.22 ±0.02 |
| $KH_2PO_4$ | 0.075 | 6.14 | 6.43 ±0.02 |

The values shown are the results of tests that were carried out at least three times.

Standard deviation (±S.D.), Control is LB broth medium without any salts supplement, -; No bacterial *E. coli* BL21 (DE3) Omp8 Rosetta's addition, +; With bacterial *E. coli* BL21 (DE3) Omp8 Rosetta's addition

**Table 2. The pH values displayed before and after *E. coli* BL21 (DE3) Omp8 Rosetta addition and cultured in LB broth prepared with various concentration of KCl at 37˚C, 200 rpm for 18 hours.**

| KCl (Molar, M) | The pH values (±S.D.) | | |
|---|---|---|---|
| | *E. coli* BL21 (DE3) Omp8 Rosetta's addition | | |
| | - | - (37˚C, 200 rpm, 18 hours) | + (37˚C, 200 rpm, 18 hours) |
| 0 | 6.30 ±0.03 | 6.34 | 6.07 ±0.02 |
| 0.025 | 6.31 ±0.01 | 6.35 | 7.79 ±0.02 |
| 0.05 | 6.30 ±0.02 | 6.30 ±0.02 | 7.85 ±0.03 |
| 0.075 | 6.30 ±0.02 | 6.31 ±0.01 | 7.96 ±0.21 |
| 0.1 | 6.30 ±0.01 | 6.32 ±0.02 | 7.98 ±0.24 |
| 0.25 | 6.25 ±0.02 | 6.26 | 7.50 ±0.10 |
| 0.5 | 6.21 ±0.01 | 6.23 | 7.77 ±0.11 |
| 0.75 | 6.20 ±0.01 | 6.21 ±0.01 | 7.30 |
| 1 | 6.18 | 6.20 | 6.18 ±0.02 |
| 1.5 | 6.18 | 6.18 | 6.16 ±0.01 |
| 2 | 6.18 | 6.18 | 6.16 ±0.01 |

The values shown are the results of tests that were carried out at least three times.

Standard deviation (±S.D.), Control is LB broth medium without any salts supplement (0 M), -; No bacterial *E. coli* BL21 (DE3) Omp8 Rosetta's addition, +; With bacterial *E. coli* BL21 (DE3) Omp8 Rosetta's addition

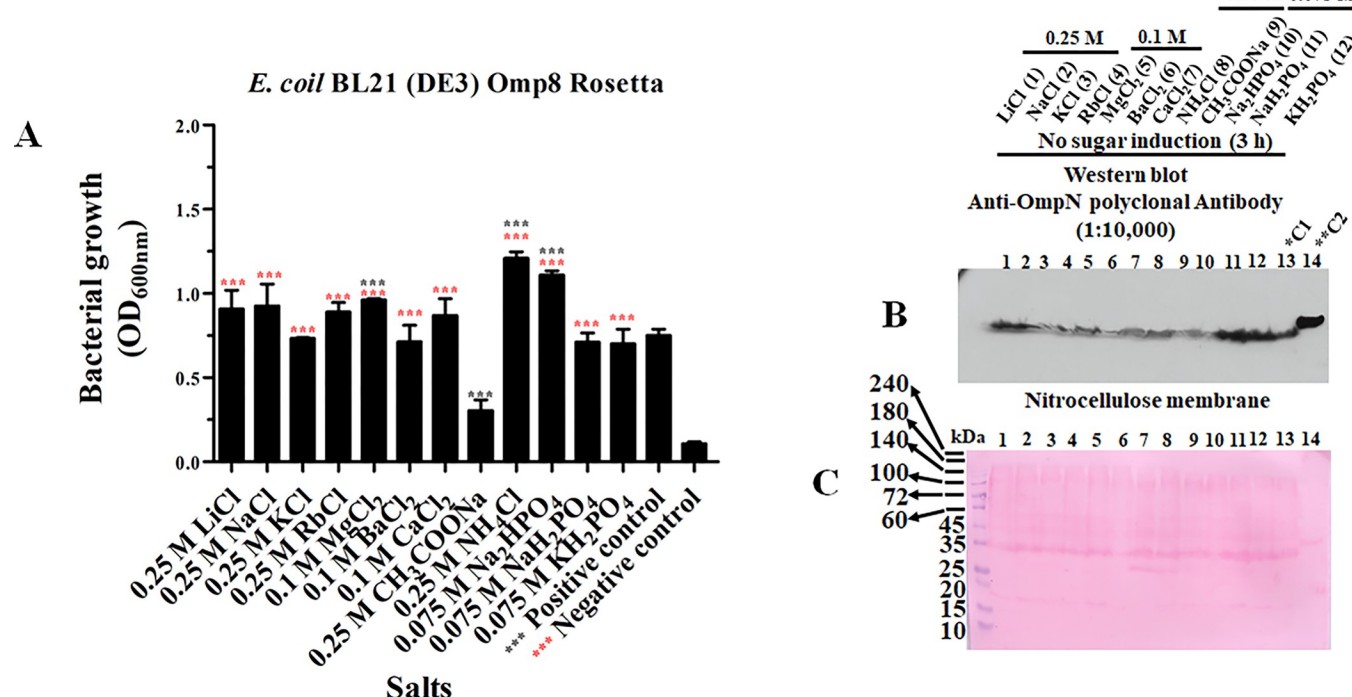

**Fig 4. OmpN expression in *E. coli* BL21 (DE3) Omp8 Rosetta strains without sugar stimulation.** This bacteria grew in LB broth medium with various salts at the optimum growth concentration (A). Crude OmpN expression was separated by 12 per cent SDS-PAGE and transferred to nitrocellulose membrane for Western blotting analysis by an anti-OmpN polyclonal antibodies (B). Ponceau S solution was used to stain the nitrocellulose membrane as an internal sample loading control (C).

for internal protein loading control (Fig 4C). The outcome revealed various levels of protein loading, confirming that OmpN protein expression level is dependent on bacterial growth. Furthermore, the OmpN protein expression was stimulated using glucose and fructose. The nitrocellulose membrane from transferred SDS-PAGE presented for glucose (Fig 5A and 5B) and fructose (Fig 5C and 5D) after being incubated with an anti-OmpN polyclonal antibody and stained with Ponceau S. OmpN expression was found to be significantly expressed for 3 hours after fructose and glucose stimulation.

Additionally, it was shown that OmpN expression in the outer membrane of Gram-negative bacteria under osmotic stress was confirmed by recombinant BOR production. Initially, the recombinant pET21b(+)/*ompN*-BOR was used to transform into the *E. coli* BL21 (DE3) Omp8 Rosetta strain, which led to the induction of recombinant BOR expression by 0.5 mM IPTG. Before IPTG induction, bacterial growth was seen at all salts tested for four hours (Fig 6A). For $CH_3COONa$, there was less bacterial growth. 12 Per cent SDS-PAGE (Fig 6B) revealed low crude OmpN extraction levels for $CH_3COONa$ in the absence of IPTG induction, which prevented anti-OmpN polyclonal antibodies from detecting OmpN. For $BaCl_2$, less OmpN expression was seen. Furthermore, significant OmpN expression was seen in the LB broth medium containing LiCl, NaCl, KCl, RbCl, and $MgCl_2$ (Fig 6C). Following the incubation of nitrocellulose membrane with anti-histidine tag polyclonal antibodies, no protein band was found. Recombinant BOR protein was only identified in positive control (Fig 6D) after the nitrocellulose membrane was stained with Ponceau S. Fig 6E shows an internal check of the protein loading.

Crude recombinant BOR protein was extracted using 2x SDS-PAGE loading buffer and boiled at 100˚C for ten minutes after being induced with 0.5 mM IPTG for three hours. The micro-centrifuged tube having a crude sample protein was then centrifuged again, and the protein was put onto two pages of 12 per cent SDS-PAGE. One was used to transfer into a nitrocellulose membrane and the other was coloured with Coomassie brilliant blue (Fig 7A). Except for $BaCl_2$, $CH_3COONa$, and $KH_2PO_4$, all salts tested included in each LB broth medium showed an OmpN protein band following incubation with anti-OmpN polyclonal antibodies (Fig 7B). The same nitrocellulose membrane was washed with mild stripping solution and used for another incubation with anti-histidine tag polyclonal antibodies (Fig 7C). After being exposed to both antibodies, the nitrocellulose membrane was stained with Ponceau S (Fig 7D).

## The pH effect of bacterial growth and recombinant OmpN protein expression on Gram-negative bacterial strain of *E. coli* BL21 (DE3) Omp8 Rosetta

The recombinant plasmid pET21b(+)/*ompN*-BOR, which caused recombinant BOR protein expression, was carried by the *E. coli* BL21 (DE3) Omp8 Rosetta bacterial strain, which was used to confirm recombinant BOR protein expression upon bacterial growth. It was decided to modify the pH value of the LB broth medium containing 0.25 M salts of either NaCl (Fig 8A) or KCl (Fig 8B) to 4, 5, 6, 7, 8, and 10. A single colony of the *E. coli* BL21 (DE3) Omp8 Rosetta bacterial strain, which carried the plasmid pET21b(+)/*ompN*-BOR, was picked up, grown for a further four hours, and then stimulated with 0.5 mM IPTG for three hours. For bacterial growth, pH values of 7 and pH 8 were shown to be the optimum pH ranges when 0.25 M NaCl and 0.25 M KCl were used, respectively. Crude recombinant BOR protein was extracted using a 2x loading buffer and boiled for 10 minutes at 100˚C and then separated into two pages of 12 per cent SDS-PAGE after being centrifuged at 13,680 x g for 10 minutes. Coomassie brilliant blue was used to stain one page, and another was transferred into a

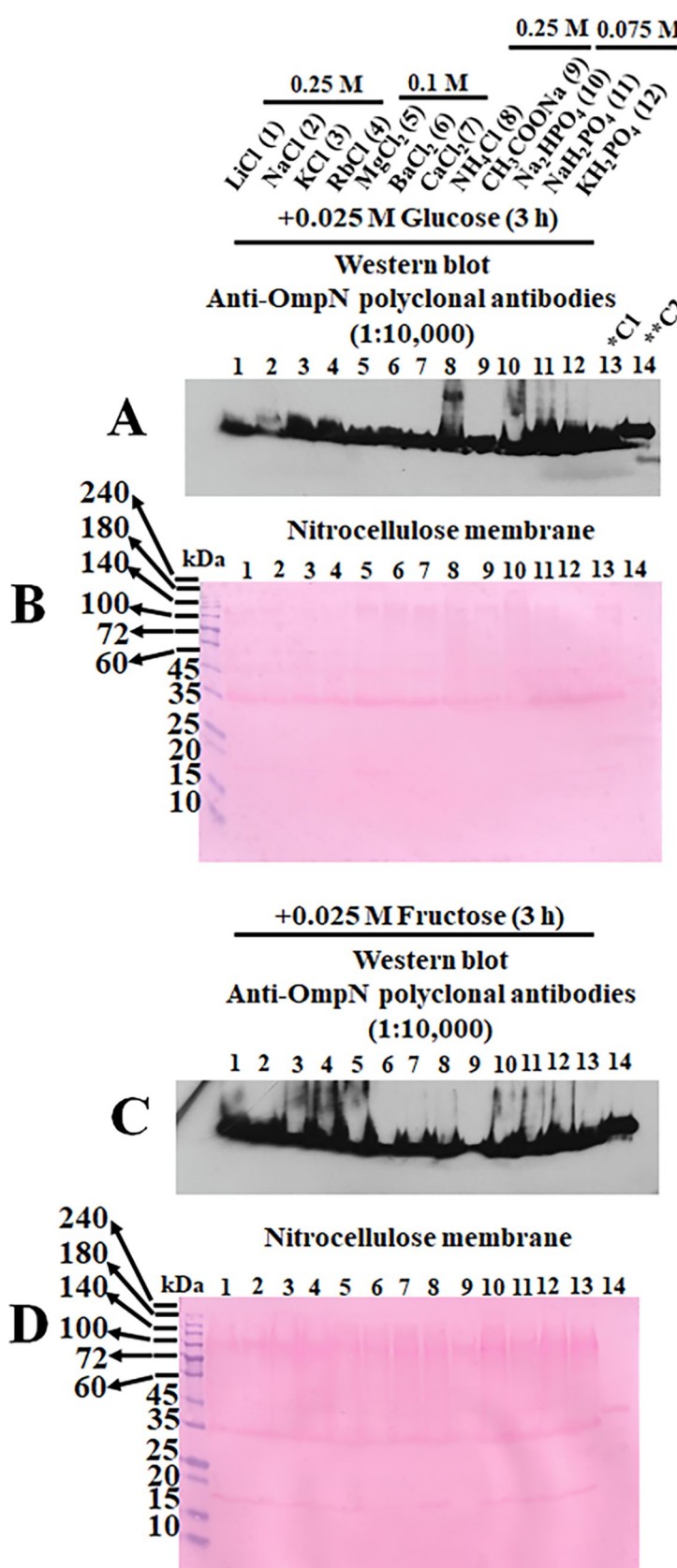

**Fig 5. _E. coli_ BL21 (DE3) Omp8 Rosetta strains of Gram-negative bacteria with sugar stimulation for OmpN expression.** Western blot using an anti-OmpN polyclonal antibodies and the nitrocellulose membrane stained with Ponceau S solution were demonstrated in the presence of glucose (A-B) and fructose (C-D).

nitrocellulose membrane. With anti-histidine tag polyclonal antibodies, both pages of the nitrocellulose membrane were incubated. The outcomes showed that, with 0.25 M NaCl, large levels of recombinant BOR protein expression were seen at pH values of 6 and 7. However, these levels were lower at pH values of 5 and 8, while pH values of 4 and pH 10 showed no protein band. A high level of recombinant BOR protein expression was seen at pH value 8 when 0.25 M KCl was present in LB broth medium, but pH values of 6, 7, and 9 showed reduced expression. There was no evidence of protein expression at pH values of 4 or 10.

## Discussion

OmpN was expressed from the target _ompN_ gene, which was initially identified in the genome of _E. coli_ K-12. OmpN was extracted and investigated in vitro using black lipid membranes (BLMs). The conductance of a completely opened channel was represented as 1.63 ±0.06 nS.

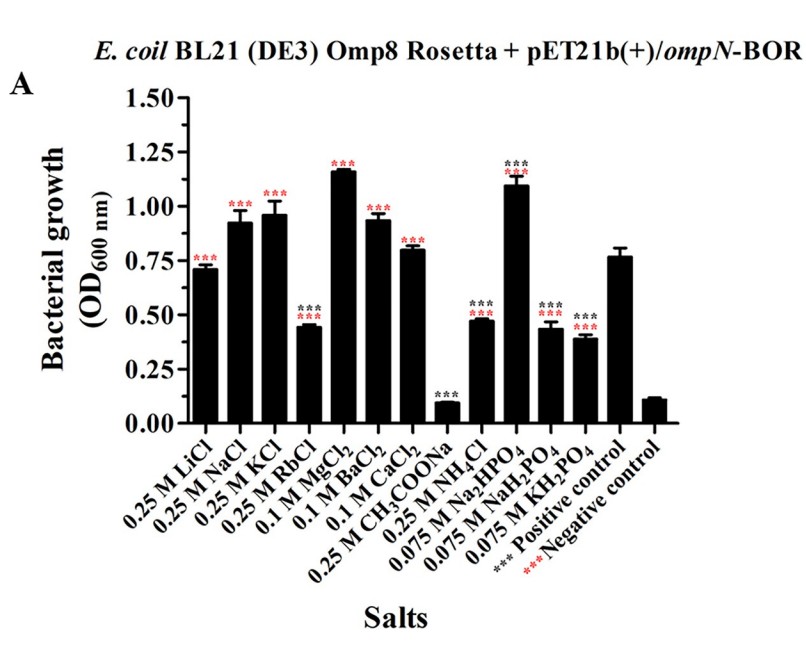
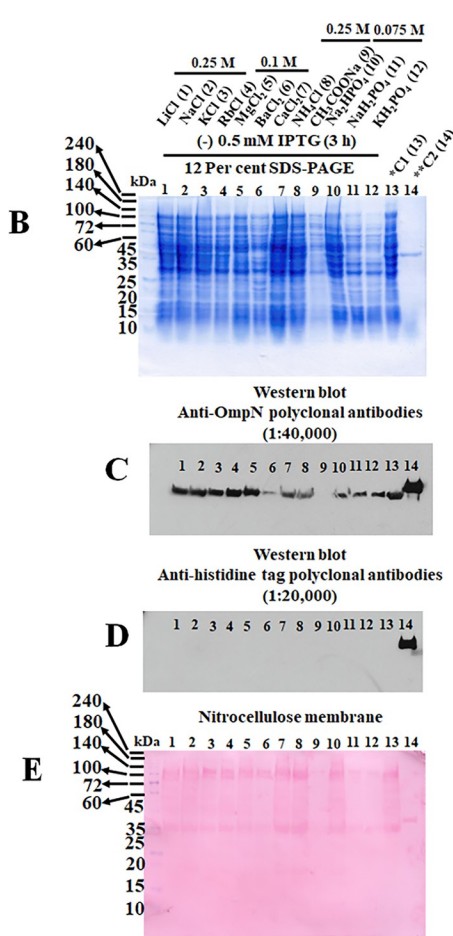

**Fig 6. The _E. coli_ BL21 (DE3) Omp8 Rosetta strain's recombinant BOR expression profile.** Without IPTG induction, the bacterial growth at optical density of 600 nm was present (A). Crude recombinant BOR protein was subsequently separated by 12 per cent SDS-PAGE and stained with Coomassie brilliant blue (B). Anti-OmpN (C) and anti-histidine tag polyclonal antibodies (D) were used to treat nitrocellulose membrane. For internal sample loading control, the nitrocellulose membrane was stained with Ponceau S solution (E).

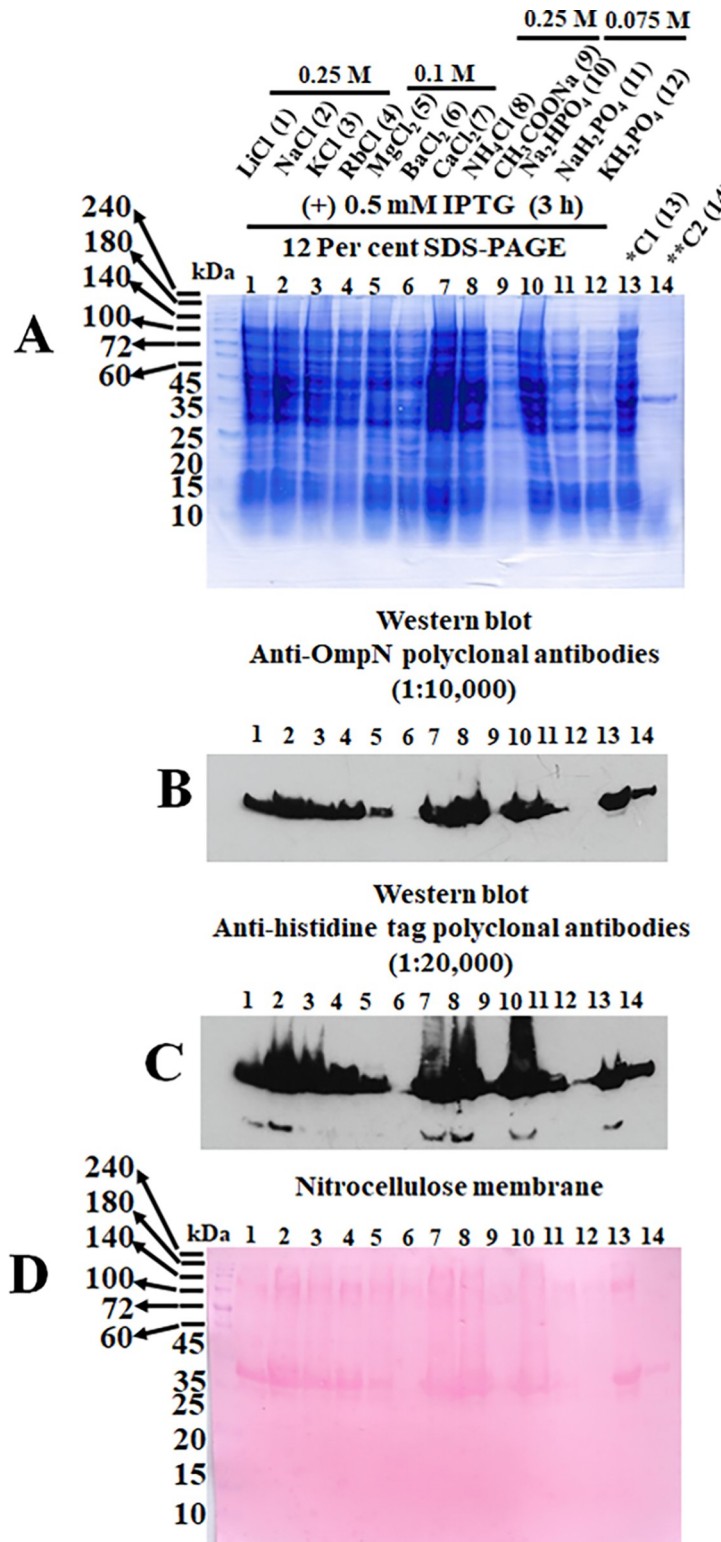

**Fig 7. Recombinant BOR protein expressed in the *E. coli* BL21 (DE3) Omp8 Rosetta strain by 0.5 mM IPTG induction for three hours.** Crude recombinant BOR was separated by 12 per cent SDS-PAGE and stained with Coomassie brilliant blue (A). The treatment of nitrocellulose membrane was carried out with anti-OmpN (B) and anti-histidine tag polyclonal antibodies (C). For an internal sample loading control, the nitrocellulose membrane was stained with Ponceau S solution (D).

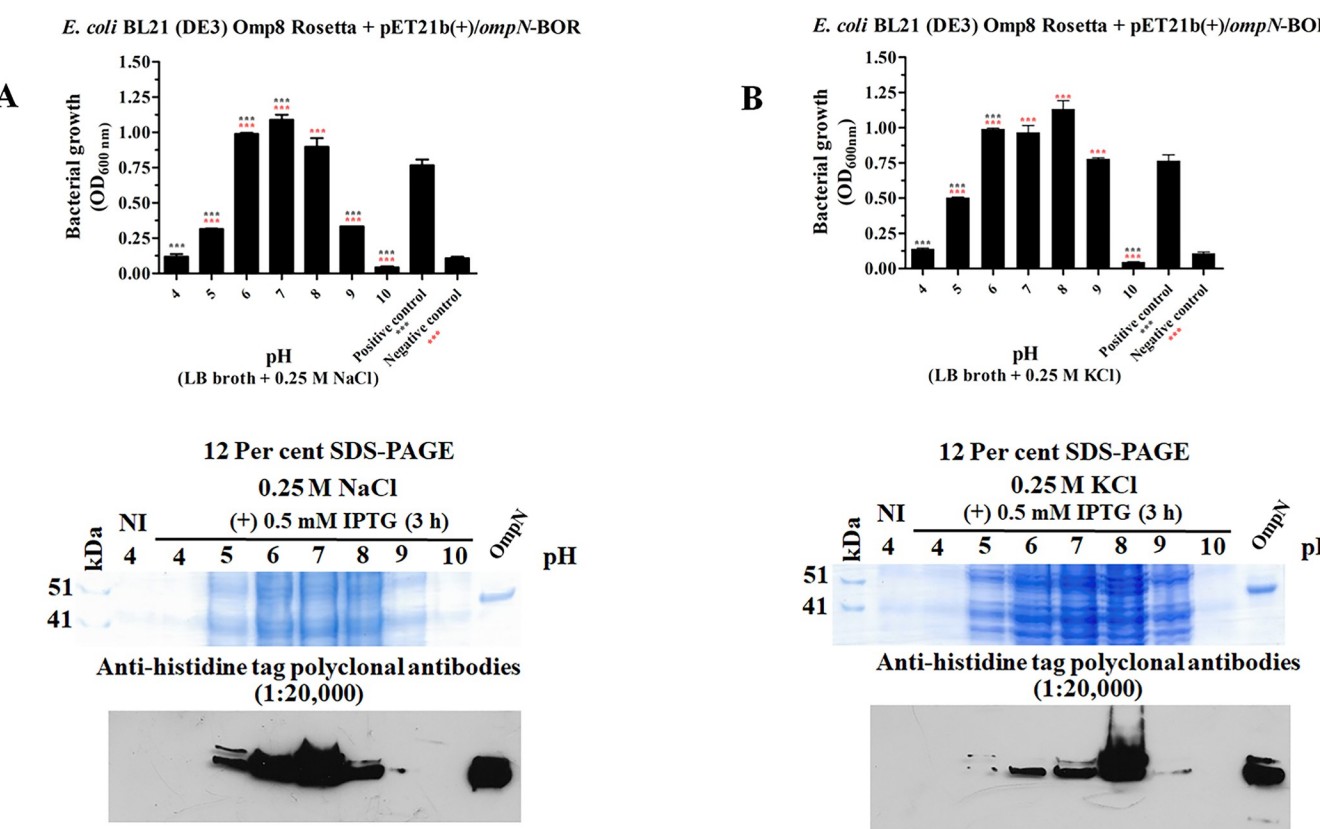

**Fig 8. Recombinant BOR protein expressed in the *E. coli* BL21 (DE3) Omp8 Rosetta strain at various pH values.** The bacterial strains harbouring the plasmid of pET21b(+)/*ompN*-BOR were grown in LB broth with 0.25 M NaCl (A) and 0.25 M KCl (B). Crude recombinant BOR protein was separated by 12 per cent SDS-PAGE and stained with Coomassie brilliant blue. Nitrocellulose membrane was treated with anti-histidine tag polyclonal antibodies.

Additionally, the pNa/pCl ratio of the OmpN channel was 4.8 ±0.8, indicating cation-selectivity [27]. Agreement with our 3D structure was achieved based on the corrected DNA sequencing of synthesized *ompN* to OmpN from *E. coli* K-12 using OmpK36 (7q3t.1.A), which shares 70.55% of its sequence [34]. It was revealed that the recombinant BOR in term of the vacuum electrostatics of its trimeric structure are presented in a 3D structure (Fig 9). Cation selectivity is the property of a channel that favours cations over anions because red has more negative than positive surface content (blue). Therefore, prior research led to the findings in this research, and we addressed bacterial proliferation at different salt concentrations in this article. The *E. coli* BL21 (DE3) Omp8 Rosetta bacterial strain was the target of our investigation since it usually expresses OmpN, a protein necessary for nutrient uptake in the outer membrane. In contrast to the lack of bacterial growth in LB broth medium without any salt supplement, our prior research showed that all salts, with the exception of CsCl, could sustain bacterial strain survival at 0.25 M. With the exception of $CaCl_2$, these bacteria can survive at 0.25 M in $MgCl_2$ and $BaCl_2$. At 0.1 M, however, it grows well. Additionally, we noticed that the *E. coli* BL21 (DE3) Omp8 Rosetta strains, with the exception of $CH_3COONa$ and $NH_4Cl$, could not grow at all on LB broth medium containing $CH_3COONH_4$. Greater evidence of bacterial growth was also detected in LB broth medium containing the other three distinct salts $Na_2HPO_4$, $NaH_2PO_4$, or $KH_2PO_4$. We demonstrated how changing the concentrations of phosphate from sodium and potassium affected the growth of the target bacterium. Like the chlorides of $MgCl_2$, $CaCl_2$, and $BaCl_2$, high bacterial growth was noted at 0.1 M, but not at 0.25 M, as was

## 3D structure of trimeric recombinant BOR protein

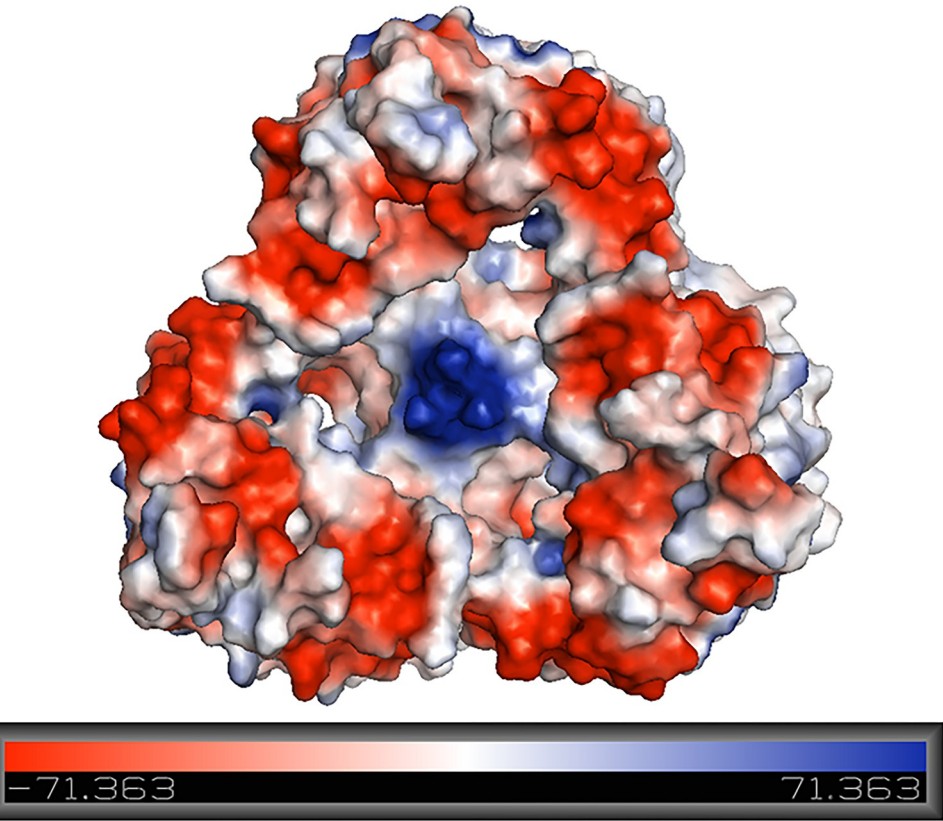

**Fig 9. The trimeric structure of recombinant BOR protein was built using an OmpK36 template (7q3t.1.A).** Vacuum electrostatics using pymol produced the surface charges, both negative and positive.

the case for the chlorides of other salts from other monovalent metal ions. Many papers have clarified that all microbial growth requires a medium with salinity stress that is produced by salt [16,35–39]. Long ago, Sherman et al. showed that *E. coli* could grow at varied pH values and NaCl concentrations when peptone was present at 1%. The outcomes demonstrated that 0.2 M at low to high concentrations of H-ion had the best influence on bacterial growth [18]. Other observations on the bioluminescence of the three luminous bacterial species under their growth conditions of two-dimensional diffusion gradients of NaCl and H$^+$ concentration. All of the *Photobacterium leiognathi*, *Photobacterium phosphoreum*, and *Vibrio fischeri* grew in the range of 0.9–3% NaCl, pH value less than 7, and temperature at 20°C, except for *Vibrio fischeri* at 15°C [40].

Bacterial proliferation led to OmpN expression. OmpN expression was detected by Western blot analysis using an anti-OmpN polyclonal antibody. It was shown to be lower without sugar induction and higher after the bacterial strain was stimulated with either 0.025 M glucose or fructose. Furthermore, OmpN expression was confirmed in the outer membrane of the *E. coli* BL21 (DE3) Omp8 Rosetta strain using recombinant OmpN expression induced by IPTG, known as BOR. After the pET21b(+)/*ompN*-BOR plasmid was used to transform into the *E. coli* BL21 (DE3) Omp8 Rosetta strain, the colony was picked up and allowed to grow. OmpN expression occurred without the need for IPTG induction, and anti-histidine polyclonal antibodies were unable to detect any recombinant BOR. When bacterial strains were cultivated in

LB broth medium that included various salts in a particular order, specifically $MgCl_2 > RbCl > KCl > NaCl > LiCl$, high levels of OmpN expression were seen. When the bacterial strain was cultivated on LB broth medium containing $CH_3COONa$, no protein band was visible. $BaCl_2$, $CH_3COONa$, and $KH_2PO_4$ in LB broth medium had no effect on OmpN expression after 3 hours of 0.5 mM IPTG induction. However, $MgCl_2$ had a reduced effect. The expression of the recombinant BOR also showed a pattern of variation with pH value. In a LB broth medium containing 0.25 M NaCl or KCl, the pH value was changed to a range of 4 to 10. Low or absent OmpN expression was a result of decreased bacterial growth in response to lower and higher pH value.

Previous research revealed that salt stress causes numerous genes to become more active [41,42]. *ompF* and *ompC* genes are two examples of genes that are upregulated and express both OmpF and OmpC [43,44]. Both OmpF and OmpC, together with PhoE, were found in *E. coli* K-12 [20]. Upregulation occurred under osmolarity to produce both outer membrane proteins. High osmolarity was favoured for the production of OmpC, whereas OmpF was produced with low osmolarity [45]. Black lipid membranes were displayed to characterise the function. Both OmpF and OmpC were extracted and purified for use in in vitro experiments. Both proteins were tested and demonstrated a fully opened channel conductance for ion transport at various salt concentrations in the buffer. The ion conductance was reported to be small at low concentrations and larger at high concentrations, as is typically found for all porins [15,46]. In in vitro investigations, CsCl was found to have a substantial ion conductance of OmpF in a series of LiCl, NaCl, and CsCl that was contained in a buffer at 0.1 M, but LiCl had a very low ion conductance [8]. Similar to our findings in vitro investigations, monovalent cations showed a significant contribution to bacterial growth while divalent cations worked as a restriction to inhibit bacterial growth. The outer membrane protein N was extensively expressed for monovalent cations, while reduced expression was reported for divalent cations, and no OmpN protein expression was observed for sodium acetate as a result of bacterial development on the LB broth medium containing salts. Following stimulation of the bacterial strain with either 0.025 M glucose or fructose, OmpN expression was shown to be overexpressed. Similar to the recombinant BOR protein, overexpression was seen with the presence of 0.5 mM IPTG for 3 hours, but no protein band was seen for barium chloride, sodium acetate, or potassium phosphate.

## Conclusion

The *E. coli* BL21 (DE3) Omp8 Rosetta strain is a gram-negative bacterial strain resistant to the drug kanamycin. This strain is typically used for producing outer membrane proteins. It lacks OmpF, OmpC, OmpA, and LamB, four of the main porins. According to earlier studies by the author, this bacterial strain produces the outer membrane protein N and *Vh*ChiP from the *chip* gene in *Vibrio harveyi* [24]. There is no evidence of the effect of osmotic stress on the other three factors such as salts, sugars, and pH value. The isolation of OmpN from *E. coli* K-12 and functional characterisation by black lipid membranes were only shown in an in vitro investigation by Prilipov et al [27]. The outcomes demonstrated the cation selectivity of its channel conductance and its preference for cation over anion. Also, the liposome swelling assay shows that the channel prefers monosugars over disugars. An earlier study enabled the author to test the notion that osmotic stress from salts, sugars, and any pH value affected OmpN expression in this bacterial strain.

The authors' finding indicated that growth in varied salt supplements in an LB broth medium had an impact on the bacterial growth of the *E. coli* BL21 (DE3) Omp8 Rosetta strain. All salts tested, except for $CaCl_2$, $CH_3COONH_4$, and $NaH_2PO_4$, had an impact on bacterial

growth at 0.25 M, whereas CsCl inhibited bacterial growth at all doses. This demonstrated that bacterial growth is dependent on salt [5,17], much as *Bacillus cereus* was dependent on NaCl [7]. To induce OmpN expression and identify it using anti-OmpN polyclonal antibodies under bacterial growth conditions, glucose and fructose were used. After the addition of either glucose or fructose for three hours, high OmpN expression was observed. Low OmpN expression was observed in the absence of sugar stimulation. Additionally, OmpN expression on the outer membrane of the *E. coli* BL21 (DE3) Omp8 Rosetta strain was verified using recombinant BOR protein. All salts, with the exception of $BaCl_2$, $CH_3COONa$, and $KH_2PO_4$, showed high levels of OmpN expression after 0.5 mM IPTG induction for 3 hours. Another observation of recombinant BOR protein expression at different pH values was that it was not detected at pH values of 4 and 10, when LB broth medium contained both NaCl and KCl. For NaCl, high OmpN expression levels were seen at pH values of 6 and 7, while KCl showed high OmpN expression levels at pH value of 8. All results lead to the conclusion that osmoregulation by the salts, sugars, and pH values studied influences bacterial growth as well as OmpN expression levels. Salt, sugar, and pH value are necessary for OmpN and recombinant BOR protein expression. This information will be used by the author to do additional research on how ions affect the conductance of the OmpN channel [8–10,14,27,46]. Additionally, the interaction of the recombinant OmpN protein with sugar or antibiotics using black lipid membranes will be investigated [24].

## Supporting information

**S1 Fig. Bacterial *E. coli* BL21 (DE3) Omp8 Rosetta strain growth at optical density 600 nm in LB broth with KCl and $CaCl_2$ at concentration of 0 to 2.0 M.** It is important to note that all of the data came from three different experiments.
(TIF)

**S1 File.**
(PPTX)

**S1 Table. The pH values of LB broth medium containing monovalent and divalent cations was present.**
(DOC)

**S2 Table. The $NH_4Cl$, $CH_3COONa$, $CH_3COONH_4$, $Na_2HPO_4$, $NaH_2PO_4$, and $KH_2PO_4$ components of LB broth medium were present, as well as the pH value.**
(DOCX)

## Acknowledgments

The authors would also like to thank the Department of Chemistry, Faculty of Engineering at the Rajamangala University of Technology Isan, Khon Kean campus and the Biochemistry-Electro-chemistry Research Unit, School of Chemistry, Institute of Science at Suranaree University of Technology for partially supporting this research.

## Author Contributions

**Conceptualization:** Watcharin Chumjan.

**Formal analysis:** Watcharin Chumjan.

**Methodology:** Watcharin Chumjan, Akira Sangchalee, Cholthicha Somwang, Nattida Mookda, Sriwannee Yaikeaw, La-or Somsakeesit.

**Writing – original draft:** Watcharin Chumjan.

**Writing – review & editing:** Watcharin Chumjan.

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
