## [Decision Letter · Decision Letter 0]

4 May 2023

PONE-D-23-08157EcOmpN expression in kanamycin antibiotic-resistant bacterial strains under osmoregulation by salts, sugars, and pHsPLOS ONE

Dear Dr. Chumjan,

Thank you for submitting your manuscript to PLOS ONE. After careful consideration, we feel that it has merit but does not fully meet PLOS ONE’s publication criteria as it currently stands. Therefore, we invite you to submit a revised version of the manuscript that addresses the points raised during the review process.

We look forward to receiving your revised manuscript.

Kind regards,

Bashir Sajo Mienda, PhD

Academic Editor

PLOS ONE

Journal Requirements:

"This research project was supported by the Rajamangala University of Technology Isan (Contract no. ENG21/65) and Thailand Science Research and Innovation (TSRI) (Contract no. FF66-P1-024). The authors would also like to thank the Department of Chemistry, Faculty of Engineering at the Rajamangala University of Technology Isan, Khon Kean campus and the Biochemistry-Electro-chemistry Research Unit, School of Chemistry, Institute of Science at Suranaree University of Technology for partially supporting this research."

Please remove any funding-related text from the manuscript and let us know how you would like to update your Funding Statement. Currently, your Funding Statement reads as follows: "no"

4. Thank you for stating the following financial disclosure: "no"

5. Thank you for stating the following in your Competing Interests section: "no" 

7. We note that you have stated that you will provide repository information for your data at acceptance. Should your manuscript be accepted for publication, we will hold it until you provide the relevant accession numbers or DOIs necessary to access your data. If you wish to make changes to your Data Availability statement, please describe these changes in your cover letter and we will update your Data Availability statement to reflect the information you provide.

8. PLOS ONE now requires that authors provide the original uncropped and unadjusted images underlying all blot or gel results reported in a submission’s figures or Supporting Information files. This policy and the journal’s other requirements for blot/gel reporting and figure preparation are described in detail at https://journals.plos.org/plosone/s/figures#loc-blot-and-gel-reporting-requirements and https://journals.plos.org/plosone/s/figures#loc-preparing-figures-from-image-files. When you submit your revised manuscript, please ensure that your figures adhere fully to these guidelines and provide the original underlying images for all blot or gel data reported in your submission. See the following link for instructions on providing the original image data: https://journals.plos.org/plosone/s/figures#loc-original-images-for-blots-and-gels. 

9. We note that you have included the phrase “data not shown” in your manuscript. Unfortunately, this does not meet our data sharing requirements. PLOS does not permit references to inaccessible data. We require that authors provide all relevant data within the paper, Supporting Information files, or in an acceptable, public repository. Please add a citation to support this phrase or upload the data that corresponds with these findings to a stable repository (such as Figshare or Dryad) and provide and URLs, DOIs, or accession numbers that may be used to access these data. Or, if the data are not a core part of the research being presented in your study, we ask that you remove the phrase that refers to these data.

10. Please include captions for your Supporting Information files at the end of your manuscript, and update any in-text citations to match accordingly. Please see our Supporting Information guidelines for more information: http://journals.plos.org/plosone/s/supporting-information. 

Reviewers' comments:

Reviewer's Responses to Questions

**Comments to the Author**

1. Is the manuscript technically sound, and do the data support the conclusions?

Reviewer #1: Partly

Reviewer #2: Partly

Reviewer #3: Partly

2. Has the statistical analysis been performed appropriately and rigorously? 

Reviewer #1: No

Reviewer #2: No

Reviewer #3: N/A

3. Have the authors made all data underlying the findings in their manuscript fully available?

Reviewer #1: No

Reviewer #2: Yes

Reviewer #3: Yes

4. Is the manuscript presented in an intelligible fashion and written in standard English?

Reviewer #1: Yes

Reviewer #2: No

Reviewer #3: Yes

5. Review Comments to the Author

Reviewer #1: Q1. The data do not fully support the conclusion because the conclusion was poorly written which is more like summary of what has been done.

Q2. Most of the result does not capture whether the analysis was significant or otherwise especially in tables and figures, indicating that can make the reader understand the validity of the data.

Q3. The data points behind means were not provided, the attached files were just like what was presented in the tables i.e the Means and standard deviations

Q4. Yes BUT there is a need for improvement in some certain sections (Indicated in the attached comment sheet) so that the manuscript can be more technically sound

Reviewer #2: This manuscript reports the results of a laboratory research investigating the growth and expression of the outer membrane protein N (OmpN) in E. coli BL21 (DE3) Omp8 Rosetta under osmotic stress conditions caused by different salts, sugars, and pHs. I recommend the issues below to be tackled.

1) The title uses ‘kanamycin antibiotic-resistant bacterial strains’ while in the manuscript only E. coli BL21 (DE3) Omp8 Rosetta is mentioned. This may cause confusion for readers expecting information about several strains. Therefore, for coherence, the authors should either provide information about other strains or revise the title to reflect the focus on only one strain.

2) ‘EcOmpN’ is not a standard protein name and should be changed to ‘OmpN’ throughout the text, as it is the correct short form for the outer membrane protein N.

3) In line 31, it should be ‘tryptone yeast extract’ instead of ‘tryptone yeast’.

4) In lines 32-33, ‘the optical density at 600 nm was evaluated’ should be changed to ‘the optical density at 600 nm was measured’.

5) In line 34, ‘all salts’ is ambiguous.

6) In line 37-38, ‘Following the presence of the plasmid pEt21b(+)/ompN-BOR in these bacterial strains, which was expressed in the recombinant BOR protein’ it's not clear what ‘these bacterial strains’ refers to. It should be clarified.

7) In line 38, ‘pEt21b(+)/ompN-BOR’ should be ‘pET21b(+)/ompN-BOR’.

8) In line 41, define the abbreviation of ‘IPTG’ when first appeared in the manuscript.

9) In line 44, ‘According to research, bacterial growth in addition to salts, sugars, and pHs influences how the EcOmpN protein is produced’ is confusing. It is unclear what research is being referred to.

10) In line 61, the authors should specify which type of bacteria was used.

11) In line 71 ‘There are variations in the genotypes’ is not necessary and can be deleted.

12) In line 90,’ Klebsiella pneumonia’ should be ‘Klebsiella pneumoniae.’

13) In line 91, ‘which increases the OmpK36’ should be ‘which increases the expression of OmpK36.’

14) In line 97, ‘To create the tryptone-medium preparation, LiCl, NaCl, KCl, RbCl, CsCl, MgCl2, CaCl2, BaCl2, Na2HPO4, NaH2PO4, and KH2PO4 were used’ is unnecessarily long and redundant.

15) In line 131, ‘BL-broth’ should be written as ‘LB broth’ as it stands for Luria Bertani broth.

16) In line 133, the chemical ‘yeast’ should be ‘yeast extract.’

17) In line 155, it is not clear what ‘left by assessment’ means.

18) In line 158, the volume of glucose and fructose added to each tube should be mentioned.

19) In line 164, instead of ‘including pH 4, 5, 6, 7, 8, 9, and 10,’ it should be changed to ‘ranging from pH 4 to 10.’

20) In line 180, the use of ‘carried’ after ‘Rosetta’ is redundant.

21) In line 193, the type of bacterial cells used should be mentioned.

22) In line 211, it is not clear which groups were compared using the t-test. The names of the groups being compared should be spell out clearly.

23) In line 214, the use of ***P <0.0001 to evaluate statistical significance is incorrect, should be ***P <0.001.

24) In lines 232-247 and figures 1A, 1B, 2A and 2B, the text and figures described the growth of bacterial strains at varying salt concentrations, however no statistical analysis (P value) is mentioned to support the conclusions.

25) Line 257, seems like other strains apart from E. coli BL21 (DE3) Omp8 Rosetta were also used. Need to be clarified to avoid confusion for the readers.

26) In lines 266-267, should specify what concentration and type of salts were used.

27) In line 267, ‘With CH3COONa, a decrease in bacterial growth was noted’ is confusing because it does not specify what was the initial bacterial growth rate, or what was the control used for comparison.

28) In line 368, ‘Additionally, we noticed that the same bacteria strains, with the exception of CH3COONa and NH4Cl, could not grow at all on LB-medium containing CH3COONH4.’ This statement is vague, in addition the correct term is ‘bacterial strains,’ not ‘bacteria strains.’

29) In line 373, ‘Like the chlorides of MgCl2, CaCl2, and BaCl2 in group 2A, high bacterial growth was noted at 0.1 M, but not at 0.25 M, as was the case for the chlorides of other salts from other metals in group 1A.’ This statement needs to be clarified; the introduction of "group 2A" and "group 1A" here is very confusing.

30) In line 380, the use of ‘other’ is redundant and should be removed.

31) In line 386, ‘which was also seen’ is redundant and can be deleted.

32) In 402, ‘Reduced bacterial growth as a result of the reduced pH led to little or no EcOmpN expression’ is unclear and should be improved.

33) In lines 406-407, ‘The two of them, together with PhoE, were found in E. coli K-12’, it is not clear whether ‘them’ refers to the OmpF and OmpC genes or the OmpF and OmpC proteins.

34) In line 417, it is not clear whose findings the authors being referred to because the statement is not properly supported by the cited references. The authors should specify whose findings they are comparing their results to.

35) In line 456, the word ‘and’ seems to be incomplete.

36) In line 464, ‘The authors finding indicated’ should be changed to ‘The authors' findings indicated’

Reviewer #3: This manuscript requires more extensive revisions because it is very limited in its scope and presentation of its results. This already starts on the title page, where the reader does not see that the authors’ university is located in Thailand. While this is merely a minor carelessness error, this style of negligence continues in the manuscript. For example, the abstract and the introduction promptly begin with a highly complex topic that is not introduced in detail. Neither the work’s relevance nor motivation becomes evident as a result.

Similarly, the abstract ends with a result without further statements about what conclusions follow from it. How this work, in turn, can be embedded in the state of research remains to be mentioned. In addition, there are, again, minor typographical errors, of which only a few can be listed. For example, Gram should be capitalized because this goes back to the name of a scientist. In the methods section, Rayong is mentioned without naming the country in which it is located. Also, typographically, there would be some things to improve. For example, the letter “x” should not be used as a substitute for the character identifying the cross product. In addition, variables should be italicized, as in the t-test, and numbers up to 12 should be written out as words. In the list of references, some terms in continuous capital letters crept in by mistake. In summary, this reviewer suggests a vast improvement of the manuscript to interest a readership beyond the very specialized studies.

6. PLOS authors have the option to publish the peer review history of their article (what does this mean?). If published, this will include your full peer review and any attached files.

Reviewer #1: **Yes: **Dr. Salim Faruk Bashir

Reviewer #2: **Yes: **Musa Hassan Muhammad

Reviewer #3: No

---

## [Author Response · Author response to Decision Letter 0]

31 May 2023

Please see in the attached file (Respond to Reviewers)

---

## [Decision Letter · Decision Letter 1]

19 Jun 2023

Outer membrane protein N expressed in Gram-negative bacterial strain of Escherichia coli BL21 (DE3) Omp8 Rosetta strains under osmoregulation by salts, sugars, and pHs

PONE-D-23-08157R1

Dear Dr. CHUMJAN,

We’re pleased to inform you that your manuscript has been judged scientifically suitable for publication and will be formally accepted for publication once it meets all outstanding technical requirements.

Kind regards,

Bashir Sajo Mienda, PhD

Academic Editor

PLOS ONE

Additional Editor Comments (optional):

Reviewers' comments:

Reviewer's Responses to Questions

**Comments to the Author**

1. If the authors have adequately addressed your comments raised in a previous round of review and you feel that this manuscript is now acceptable for publication, you may indicate that here to bypass the “Comments to the Author” section, enter your conflict of interest statement in the “Confidential to Editor” section, and submit your "Accept" recommendation.

Reviewer #1: All comments have been addressed

Reviewer #2: (No Response)

2. Is the manuscript technically sound, and do the data support the conclusions?

Reviewer #1: Yes

Reviewer #2: Yes

3. Has the statistical analysis been performed appropriately and rigorously? 

Reviewer #1: Yes

Reviewer #2: Yes

4. Have the authors made all data underlying the findings in their manuscript fully available?

Reviewer #1: Yes

Reviewer #2: Yes

5. Is the manuscript presented in an intelligible fashion and written in standard English?

Reviewer #1: Yes

Reviewer #2: (No Response)

6. Review Comments to the Author

Reviewer #1: (No Response)

Reviewer #2: This manuscript reports the results of a laboratory research investigating the growth and expression of the outer membrane protein N (OmpN) in E. coli BL21 (DE3) Omp8 Rosetta under osmotic stress conditions caused by different salts, sugars, and pHs. I recommend that the article can be accepted if the following minor revisions are addressed.

In line 548, ‘’perceived’’ should be changed to ‘’observed’’

In line 552, ‘’different pHs’’ should be changed to ‘’different pH values.’’

In line 553, ‘’it was not present’’ should be changed to ‘’it was not detected.’’

In line 553, ‘’pHs 4 and 10’’ should be changed to ‘’pH values of 4 and 10’’, and the term ‘’pH’’ should be replaced with ‘’pH value’’ throughout the manuscript.

In line 553, ‘’included both NaCl and KCl in the LB broth medium’’ should be changed to ‘’when LB broth medium contained both NaCl and KCl.’’

In line 554, ‘’while KCl showed high OmpN expression levels’’ should be replaced with ‘’while high levels of OmpN expression were observed in the presence of KCl.’’

7. PLOS authors have the option to publish the peer review history of their article (what does this mean?). If published, this will include your full peer review and any attached files.

Reviewer #1: **Yes: **Dr. Salim Faruk Bashir

Reviewer #2: **Yes: **Musa Hassan Muhammad

---

## [Editor Report · Acceptance letter]

26 Jul 2023

PONE-D-23-08157R1 

Outer membrane protein N expressed in Gram-negative bacterial strain of *Escherichia coli* BL21 (DE3) Omp8 Rosetta strains under osmoregulation by salts, sugars, and pHs 

Dear Dr. Chumjan:

I'm pleased to inform you that your manuscript has been deemed suitable for publication in PLOS ONE. Congratulations! Your manuscript is now with our production department. 

Kind regards, 

on behalf of

Dr. Bashir Sajo Mienda 

Academic Editor

PLOS ONE